# Comparison of Conformations and Interactions with Nicotinic Acetylcholine Receptors for *E. coli*-Produced and Synthetic Three-Finger Protein SLURP-1

**DOI:** 10.3390/ijms242316950

**Published:** 2023-11-29

**Authors:** Vladimir Kost, Dmitry Sukhov, Igor Ivanov, Igor Kasheverov, Lucy Ojomoko, Irina Shelukhina, Vera Mozhaeva, Denis Kudryavtsev, Alexey Feofanov, Anastasia Ignatova, Yuri Utkin, Victor Tsetlin

**Affiliations:** 1Shemyakin-Ovchinnikov Institute of Bioorganic Chemistry of the Russian Academy of Sciences, 117997 Moscow, Russia; goron.dekar@gmail.com (V.K.); sukhovdim@yandex.ru (D.S.); chai.mail0@gmail.com (I.I.); iekash@ibch.ru (I.K.); lucy.ojomoko@gmail.com (L.O.); shelukhina.iv@yandex.ru (I.S.); 1996-racer@mail.ru (V.M.); kudryavtsev@ibch.ru (D.K.); avfeofanov@yandex.ru (A.F.); utkin@mx.ibch.ru (Y.U.); 2Prokhorov General Physics Institute of the Russian Academy of Sciences, 119991 Moscow, Russia; 3Department of Biology and General Genetics, I.M. Sechenov First Moscow State Medical University, 8, bldg. 2 Trubetskaya Str., 119048 Moscow, Russia

**Keywords:** Ly6/uPAR proteins, SLURP-1, nicotinic acetylcholine receptors, recombinant protein, synthetic protein, Raman spectroscopy, CD spectroscopy, NMR, radioligand assay, computer modeling

## Abstract

SLURP-1 is a three-finger human protein targeting nicotinic acetylcholine receptors (nAChRs). The recombinant forms of SLURP-1 produced in *E. coli* differ in added fusion fragments and in activity. The closest in sequence to the naturally occurring SLURP-1 is the recombinant rSLURP-1, differing by only one additional N-terminal Met residue. sSLURP-1 can be prepared by peptide synthesis and its amino acid sequence is identical to that of the natural protein. In view of recent NMR analysis of the conformational mobility of rSLURP-1 and cryo-electron microscopy structures of complexes of α-bungarotoxin (a three-finger snake venom protein) with *Torpedo californica* and α7 nAChRs, we compared conformations of sSLURP-1 and rSLURP-1 by Raman spectroscopy and CD-controlled thermal denaturation, analyzed their competition with α-bungarotoxin for binding to the above-mentioned nAChRs, compared the respective receptor complexes with computer modeling and compared their inhibitory potency on the α9α10 nAChR. The CD revealed a higher thermostability of sSLURP-1; some differences between sSLURP-1 and rSLURP-1 were observed in the regions of disulfides and tyrosine residues by Raman spectroscopy, but in binding, computer modeling and electrophysiology, the proteins were similar. Thus, sSLURP-1 and rSLURP-1 with only one additional Met residue appear close in structure and functional characteristics, being appropriate for research on nAChRs.

## 1. Introduction

The proteins of the Ly6/uPAR family, which generally have the same spatial structure as that of three-finger snake venom neurotoxins, have been known for a long time to be present in the immune system of mammals [1], but only at the end of the last century were these proteins found in the mammalian brain and shown to interact with nicotinic acetylcholine receptors (nAChRs) [2]. Among the first such proteins are Lynx1 and SLURP-1 [2,3], the former being attached to the cell membrane by the glycosylphosphatidylinositol (GPI) tail, while SLURP-1 is secreted, lacks the GPI anchor and, thus, is closer to its structural homologs, three-finger neurotoxins. These proteins influence the assembly of nAChRs and their different functions (see recent papers [4,5,6,7]) and, thereby, are considered as possible leads to drugs [4,8,9]. Most of the respective research on GPI-containing forms has been conducted by supporting or suppressing their expression in the mammalian brain and checking the effects on nAChRs (e.g., [10,11]. As an individual protein, Lynx1 was obtained in a water-soluble form (ws-Lynx1) lacking the GPI tail; its ^1^H-NMR spatial structure has been determined and inhibition of the muscle-type and α7 nAChRs has been demonstrated [12]. The inhibition of the α7 nAChR by ws-Lynx1 was similar to that of the whole-size Lynx1 protein, but in the literature, there are also examples showing some differences in the activity of whole-size and GPI-lacking Lynx1 forms [13]. There is no such problem concerning SLURP-1, but in SLURP-1 samples heterologously produced in several laboratories, there were different added fusion portions, resulting, in several cases, in opposite effects on nAChR activity [14,15,16,17]. Most close to the naturally occurring protein is recombinant SLURP-1 with only one additional N-terminal Met residue [16], designated here as rSLURP-1. The diversity of the physiological roles of SLURP-1 found in keratinocytes and immune and cancer cells has been recently reviewed [6,17]. Mutations in the gene encoding SLURP-1 were detected in patients with Mal de Meleda, a rare autosomal recessive skin disorder characterized by transgressive palmoplantar keratoderma [18,19,20]; some mutations located in loops I and II and the head of SLURP-1 may potentially drive the disease [21]. The immunomodulatory action of SLURP-1 was shown for T cells, blood mononuclear leukocytes, dendritic cells, mast cells and macrophages; SLURP-1 also inhibits the growth of multiple cancer cell lines (reviewed in [6,17]).

In this article, we compare rSLURP-1 with synthetic sSLURP-1, which is made by total peptide synthesis [22] and identical in its amino acid sequence to the naturally occurring human protein. Our work was in part stimulated by a recent publication [23] wherein detailed ^1^H-NMR studies revealed in rSLURP-1 the presence of two isomers at the Tyr39–Pro40 bond and demonstrated the conformational mobility of this protein. Additional stimuli comprise two recent cryo-electron microscopy structures of α-bungarotoxin complexes with the muscle-type *Torpedo californica* and with neuronal α7 nAChRs [24,25], which appear to allow one, with computer modeling, to get an idea of SLURP-1 binding to these targets. Here, we compared sSLURP-1 and rSLURP-1 by spectroscopic approaches, analyzed their competition with radioactive α-bungarotoxin for binding to the muscle-type *T. californica* and to human neuronal α7 nAChRs, tested their inhibitory potency against α9α10 nAChR by electrophysiology and also compared, by computer modeling, the binding modes of sSLURP-1 and rSLURP-1 in light of the above-mentioned cryo-electron microscopy structures. In general, our results illustrate the similarity of the two SLURP-1 forms, with a slightly higher thermostability being attributed to sSLURP-1, and show that synthetic sSLURP-1 and *E. coli*-expressed rSLURP with only one additional N-terminal Met residue are adequate models for studying the function and possible practical applications of SLURP-1, a human endogenous three-finger protein.

## 2. Results

### 2.1. Preparation of rSLURP-1 and sSLURP-1

rSLURP-1 was produced by heterologous expression of the synthetic gene in *E. coli* cells. The protein was expressed in the form of inclusion bodies from which it was extracted with buffer-containing urea and dithiothreitol (DTT) as a reducing agent. The extracted protein was purified by ion exchange chromatography in the presence of the reducing agent followed by reversed-phase HPLC under strongly acidic conditions which prevented the formation of disulfide bonds. For the refolding of both proteins, we used the conditions applied in the case of sSLURP-1 refolding [22]; the reaction was left at 4 °C for 3 days. These conditions did not differ strongly from those used earlier [16] for rSLURP-1. The renatured proteins were purified by reversed-phase HPLC, after which they showed quite similar chromatographic profiles and were analyzed by mass spectrometry. The mass spectra of sSLURP-1 and rSLURP-1 corresponded to proteins with fully formed disulfide bonds, the mass of the latter being 131.2 Da larger, the difference being equal to the molecular mass of methionine residue. The proteins obtained thusly were used for further analysis.

### 2.2. Analysis of rSLURP-1 and sSLURP-1 Competition with Iodinated α-Bungarotoxin for Binding to the Torpedo californica and Human α7 nAChRs

The capability of rSLURP-1 and sSLURP-1 to interact with the muscle-type nAChR from the electric organ of the *T. californica* ray and with the human neuronal α7 nAChR in the GH4C1 cell line was evaluated by a radioligand assay in competition with [^125^I]-labeled α-bungarotoxin for binding to these targets. The dose–response dependence for both proteins in displacing bound α-bungarotoxin (Figure 1) allowed us to evaluate their affinities to the *Torpedo* receptor.

The mean IC_50_ values calculated from four independent experiments with four batches of rSLURP-1 and sSLURP-1 were 9.55 (2.3–13.1) μM and 3.92 (0.89–5.95) μM, respectively. 

The affinities of both proteins to the neuronal α7 nAChR were significantly lower: at a concentration of 30 μM, sSLURP-1 displaced the bound radioligand by 50%, and rSLURP-1displaced the bound radioligand by only 20% (Figure 1).

### 2.3. Electrophysiological Analysis of Inhibition of α9α10 nAChR by rSLURP-1 and sSLURP-1

Electrophysiological studies showed that sSLURP-1 and rSLURP-1 forms have an almost equal inhibitory activity against the rat α9α10 nAChR: at a 25 µM concentration, they decreased the Ach (500 µM)-induced response in *Xenopus laevis* oocytes by 38.5 and 35.5%, respectively (Figure 2). At 1–10 µM concentrations, both rSLURP1 (Figure 2) and sSLURP1 [22] almost lost their inhibitory potency toward the rat α9α10 nAChR, although α-conotoxin RgIA successfully inhibited this receptor at 0.17 µM (Figure 2). Earlier sSLURP-1 studies demonstrated an almost identical inhibitory potency on rat and human α9α10 nAChRs [22]. 

### 2.4. Circular Dichroism Analysis of sSLURP-1 and rSLURP-1 Thermostability

To compare the conformational stability of both SLURP-1 samples, we used CD spectroscopy. The CD spectra recorded at temperatures ranging from 20 to 90 °C show that the β-structure is the main element of secondary structure in these proteins (Figure 3, Table 1). In both proteins, the β-structure and β-turns comprise about 60% of all secondary structure elements. sSLURP-1 appears to be slightly more ordered: at practically all temperatures, it contains more β-structure and less unordered structure than rSLURP-1. The structures of both proteins are very stable and substantial changes are observed only at 90 °C (Figure 3). Interestingly, at this temperature, the content of unordered structure decreases in both proteins. The difference in the content of secondary structure elements between the two proteins becomes more evident at high temperatures. Thus, at 90 °C, in sSLURP-1, the percentage of β-structure increases compared to that at 20 °C, while in rSLURP-1, an increase in the percentage of α-helix is observed (Table 1). 

### 2.5. Raman Spectroscopy of rSLURP-1 and sSLURP-1

There are only a few examples of Raman spectroscopy’s application to three-finger snake neurotoxins [26,27]. Recently, we applied Raman spectroscopy in combination with Principal Component Analysis (PCA) and clustering methods for the investigation of three-finger neurotoxins and α-conotoxins interacting with nAChRs and demonstrated that PCA can be successfully applied to identify structural differences and similarity between the studied toxins [28]. In the present study, we used a similar approach and compared sSLURP-1 and rSLURP-1 by Raman spectroscopy (Figure 4).

Below, we consider the most informative regions of the obtained Raman spectra (Figure 4) of both sSLURP-1 and rSLURP-1 reflecting the individual structural features of proteins.

We considered the spectral regions that contained information about protein structures, namely the “S–S region” (490–550 cm^−1^) (Figure 5a), which is used to characterize the geometry of C–C–S–S–C–C bonds, and the 810–870 cm^−1^ region (Figure 5b), which contains the so-called tyrosine doublet and is used to determine the microenvironment of tyrosine residues, as well as Amide III (1230–1280 cm^−1^) and Amide I (1640–1690 cm^−1^) bands (Figure 6a,b, respectively), which contain information about the proteins’ secondary structure (see, for examples, [29,30]).

Analysis of the “S–S region” allowed us to establish that the gauche–gauche–gauche conformation of C–C–S–S–C–C prevails in the structure of both samples, since the maximum of the main peak is localized near 510 cm^−1^. However, for sSLURP-1, a small contribution of gauche–gauche–trans conformation of C–C–S–S–C–C is noticeable, as evidenced by the appearance of a shoulder at about 525 cm^−1^ (Figure 5a). This may be due to the fact that a certain percentage of sSLURP-1 molecules have this altered conformation of C–C–S–S–C–C. Since the spectra were recorded from dry samples, we can exclude dynamic processes, in particular, transitions between conformations (during recording spectra), in this case. It is interesting to note that in the Raman spectrum of the three-loop α-cobratoxin obtained earlier, a contribution from the gauche–gauche–trans conformation was also observed. This may indicate greater similarity in the geometry of the sSLURP-1 and α-cobratoxin loops. 

Analysis of the tyrosine doublet (Figure 5b) suggests that the tyrosine residues in both samples have a predominantly hydrophobic environment (since the main peak is localized around 830 cm^−1^), but it should be noted that, in contrast to sSLURP-1, some portion of tyrosine residues in the rSLURP-1 molecule have a hydrophilic environment (additional maximum about 850 cm^−1^). This probably indicates a greater exposure of some tyrosine residues. The presence of energetically unfavorable tyrosine residues in a hydrophilic environment in rSLURP-1 may indicate a lesser stability of this SLURP-1 variant. However, other factors, such as a different, more energetically favorable disulfide geometry (without a significant contribution of the gauche–gauche–trans conformation in C–C–S–S–C–C bonds), can balance the energetically unfavorable contribution of tyrosine residues in a hydrophilic environment.

Analysis of the secondary structure of the samples does not reveal significant differences: both show the presence of a β-structure, as evidenced by the position of the main peaks at about 1240 cm^−1^ for Amide III and 1670 cm^−1^ for Amide I (Figure 6a,b).

As described in [28], PCA of Raman spectra allowed us to extract the most significant features to distinguish various toxins based on their structures (amino acid content, secondary structure, disulfides, etc.) and represent them in an abstract low-dimensional space in the convenient form of a 2D plot (Figure 7). As can be seen from the corresponding loading plot (loadings spectra are shown on Appendix A), various regions (marker bands), characterizing both vibrations of the protein backbone and sidechains, contribute to the discrimination of various toxins. So, it is difficult to highlight any specific region (or several regions) responsible for this clustering. Raman spectra of rSLURP-1, sSLURP-1 and several subtype-selective nAChR-targeted neurotoxins were compared using PCA. The performed analysis showed a close relation of both rSLURP-1 and sSLURP-1 structures to other three-finger proteins, such as α-cobratoxin (Ctx) and short-type α-neurotoxin NT2, but not to other nAChR subtype-selective peptides, namely azemiopsin (Az) and α-conotoxins (SIA, RL-PnIA, M1 and M2) (Figure 7).

### 2.6. Computer Modeling of rSLURP-1 and sSLURP-1 Binding to Torpedo californica and Human α7 nAChRs

Two approaches to rSLURP-1’s docking to muscle-type and α7 nAChRs were used. First, based on the data about competition with the radioligand, docking to the orthosteric binding sites of muscle-type and α7 nAChRs was performed on the Rosetta ToxDock server [31]. 

Binding poses of rSLURP-1 at these orthosteric sites resemble the binding pose of α-bungarotoxin (Figure 8a,b). Docking of rSLURP-1 to the α7 nAChR orthosteric site revealed a putative structure similar to the structure of α-bungarotoxin (red) in complex with this nAChR subtype. Loop II of rSLURP-1 is positioned under loop C of the principal face of the orthosteric binding site of the α7 subunit. Binding energy estimated by the Rosetta scoring function shows preferential interaction of rSLURP-1 with the muscle-type nAChR compared to the α7 receptor (Figure 8c). This correlates well with radioligand competition data. To explore possible binding modes of rSLURP-1 to the α7 nAChR outside the classical orthosteric binding pocket, we performed un-guided macromolecular docking using the HDOCK server [32].

As shown in Figure 8d,e, several putative binding sites were identified outside the orthosteric site. Three main types of sites can be concluded from these results: (1) at the top of vestibulum, (2) at the intersubunit interface near loop B and the α1–β1 linker and (3) below loop C. It is worth noting that the binding pose at the top of the vestibulum part of the receptor (see pose 1 on Figure 8d,e) demonstrated a possible involvement of the Met 1 residue in binding (Figure 8f). 

## 3. Discussion

Three-finger proteins from snake venom continue to play an important role together with α-conotoxins (neurotoxic peptides from marine snails) as sophisticated pharmacological tools for the whole set of heteromeric and homomeric nAChRs of the brain and immune system and of different tissues (see reviews [33,34]). In recent years, much attention has been paid to the nAChR-targeting TFPs of diverse organisms, from insects to humans (see reviews [4,5,6,35]). Some of them, like human Lynx1, are membrane-attached by the glycosylphosphatidylinositol (GPI) anchor, while others like SLURP-1 are secreted. The advantage of such proteins as Lynx1 or SLURP-1, in comparison with such well-recognized tools in nAChR research as α-neurotoxins and α-conotoxins, is their lack of toxicity. Moreover, these proteins are present in the mammal organisms and regulate some important functions, which allows one to consider them and their derivatives as possible drug leads.

The effects of GPI-anchored proteins on nAChRs have been analyzed using their enhanced or knock-out expression in organisms but, to date, none of them have been studied as an individual isolated protein. The first step in this direction was the heterologous expression in *E. coli* of a water-soluble Lynx1 (ws-Lynx1), having a total three-finger moiety of the naturally occurring protein but devoid of the GPI anchor [12]. Its ^1^H-NMR structure was determined and the activity was characterized by testing the competition with radioiodinated α-bungarotoxin for binding to the muscle-type *Torpedo californica* and human neuronal α7 nAChRs, as well as by analyzing the inhibition of ion currents in the latter [12]. The ^1^H-NMR structure of this protein was the first experimental proof that a human Ly6/uPAR protein targeting nAChRs indeed has a three-finger folding similar to that of snake venom neurotoxins. It was later indicated that the expression in mice of Lynx1 with or without a GPI tail results in different behavioral modes [13], but possible effects of the additional N-terminal Met residue in ws-Lynx1 due to the expression in *E. coli* were not discussed. With such secreted proteins as SLURP-1 expressed in *E. coli*, there was no question of a principal difference from the naturally occurring protein, although, in some cases, the added fused portions were quite large. The closest to the native product was rSLURP-1, bearing only one additional N-terminal Met residue, its ^1^H-NMR structure being almost identical to that of synthetic sSLURP-1 and having an amino acid sequence identical to that of the native protein [16,22]. 

These two proteins were antagonists of nAChRs, but some differences between them in the competition with radioactive α-bungarotoxin were registered; in addition, the inhibitory action of sSLURP-1 was shown for a wider spectrum of neuronal nAChRs, including heteropentameric ones [22]. In the present study, we continued the comparison of sSLURP-1 and rSLURP-1 by additional spectroscopic approaches, radioligand analysis, electrophysiology and computer modeling. Raman spectroscopy (Figure 4, Figure 5, Figure 6 and Figure 7) revealed, in general, a similarity in the spatial organization of these two proteins reflected in the similarity of their Amide I and Amide III bands, but small differences between sSLURP-1 and rSLURP-1 were detected in the S–S region, and the most pronounced difference was in the tyrosine doublet region, indicating disposition of some tyrosine residue(s) of rSLURP-1 in a more hydrophilic environment.

Analysis of thermodenaturation by CD spectroscopy (Figure 3, Table 1) also confirmed the similarity of sSLURP-1 and rSLURP-1 but revealed that changes in the secondary structure upon heating are different in these two samples: synthetic sSLURP-1, not having an additional Met residue and, thus, being closer to the native sample, has a higher thermostability. Interestingly, this property appears to agree with the Raman spectroscopy data on the environment of Tyr residues (Figure 5b) and Amide I and III bands (Figure 6).

In view of the recently published cryo-electron microscopy structures of α-bungarotoxin complexes with *Torpedo californica* and α7 nAChRs [24,25], we compared in more detail the interactions of sSLURP-1 and rSLURP-1 with these nAChR subtypes by their competition with radioiodinated α-bungarotoxin and also by computer modeling of the respective receptor complexes. Figure 1 shows that both samples inhibit, at a micromolar concentration, the radioligand binding to the *T. californica* nAChR, with sSLURP-1 being more efficient; sSLURP-1 is also slightly more efficient for the human α7 nAChR, although the inhibition of the α-bungarotoxin binding to the orthosteric site of this nAChR subtype by both SLURP-1 forms is very weak.

In our electrophysiological study (Figure 2), it has been shown for the first time that rSLURP-1 is able to inhibit a neuronal heteromeric α9α10 nAChR as effectively as sSLURP-1 [22]. Although our results demonstrate a similarity of properties between the protein (sSLURP-1) having the sequence of the native SLURP-1 and the protein (rSLURP-1) produced in *E. coli* and having one additional Met residue, we refrain from drawing conclusions about other SLURP-1 forms with the extended fusion portions as described in the literature. Here, it is appropriate to mention that a comparison of the activities with the naturally occurring form isolated from *Naja kaouthia* cobra venom and with the protein with an additional N-terminal Met residue produced in *E. coli* was performed for the weak toxin WTX, which belongs to the group of non-conventional toxins, has the same disposition of five disulfides as SLURP-1 and acts both on nAChRs and muscarinic acetylcholine receptors (mAChR) [36]. Having similar spatial structures, they acted similarly on nAChRs, but pronounced differences were detected towards mAChRs [36].

Modeling of binding to *T. californica* and human α7 nAChRs was performed using the spatial structure of rSLURP-1 available from recently published NMR data [23]. We found that the N-terminal Met residue was quite far from the orthosteric binding sites in these receptors, which prompted us to think that the binding mode of sSLURP-1 to this site should be similar to that of rSLURP-1. In general, they occupy the same position at the muscle-type nAChR (Figure 8) as was found in their competition with α-bungarotoxin; by modeling the binding of sSLURP-1 (using the coordinates of rSLURP-1), we could not define any additional contacts that would explain a slightly higher efficiency of interaction with the *T. californica* nAChR for sSLURP-1, as compared to rSLURP-1. A noticeably weaker inhibition of α-bungarotoxin attachment to the α7 nAChR by both SLURP-1 forms is not surprising because it has been shown that rSLURP-1 predominantly attacks this receptor subtype by binding at some allosteric sites [16]. Interestingly, un-guided docking of rSLURP-1 to the extracellular domain of the α7 nAChR revealed several alternative binding sites, and it should be noted that the allosteric binding sites were previously detected for low-molecular-weight compounds [37,38]. It should also be noted that although we have discussed the inhibitory activity of SLURP-1 via orthosteric or allosteric binding sites in nAChRs, the data in the literature suggest that, in certain cancer cell lines, rSLURP-1 acts not only on the α7 nAChR but also targets some receptors not belonging to the family of Cys-loop ligand-gated ion channels, opening new lines to potential drugs [9,39]. Thus, a detailed analysis of such Ly6/uPAR proteins as SLURP-1 and Lynx1 is worthy of continuation. Possible applications may lie in their synthetic peptide fragments as was shown for a peptide fragment of Lynx1, which preserved the spatial structure of the Lynx1 central loop II and bound to the *T. californica* nAChR as efficiently as ws-Lynx1 itself [40]. Intravenous injection of a synthetic peptide mimicking SLURP-1’s loop I (Oncotag) suppressed tumor growth and metastasis in a xenograft mice model of epidermoid carcinoma in a similar way to rSLURP-1 [41].

## 4. Materials and Methods

### 4.1. Reagents

Isopropyl β-d-1-thiogalactopyranoside (IPTG) was obtained from Thermo Fisher Scientific (Waltham, MA, USA) and tris(2-carboxyethyl)phosphine (TCEP) was obtained from Merck (Darmstadt, Germany). All other reagents were obtained from local suppliers with analytical-grade or higher purity.

### 4.2. Preparation and Purification of rSLURP-1 and sSLURP-1

#### 4.2.1. Construction of the Expression Plasmid for rSLURP-1

The human SLURP-1 gene (23-103, UniProtKB SLUR1_HUMAN) was optimized for production in *E. coli* and synthesized by Evrogen (Moscow, Russia). The synthetic gene was cloned into the expression vector pET22b. The structure of the obtained plasmid was confirmed by sequencing.

#### 4.2.2. Bacterial Production of rSLURP-1

The target gene was expressed in *E. coli* BL21 (DE3) cells. *E. coli* culture was inoculated overnight into 20 mL of LB medium containing ampicillin at a concentration of 100 mg/L. An overnight culture was introduced into 2 l of LB medium and the mixture was incubated at 37 °C on a shaker (180 rmp) for 2–3 h until optical density at 600 nm reached 0.4–0.6. Then, isopropyl β-d-1-thiogalactopyranoside (IPTG) was added up to a concentration of 0.5 mM and the cells were cultivated overnight at 34 °C. After that, the cells were pelleted by centrifugation for 10 min at 5000 rpm using a rotor JA-15. The wet cell pellets (7.5 g) were resuspended in 80 mL of 50 mM Tris-HCl buffer (pH 7.6), containing 500 mM NaCl, 10 mM EDTA and 1 mM PMSF and sonicated for 20 min on ice using a Soniprep 150 ultrasonic disintegrator (MSE Centrifuges Limited, Heathfield, UK) equipped with a large probe assy. The mixture obtained was centrifuged for 20 at 15,000 rpm and the supernatant was discarded. The pellet was resuspended in 50 mL of 50 mM Tris-HCl (pH 7.6) containing 10 mM EDTA and 2 M urea, stirred for 10 min, sonicated for 2 min as above and stirred for 10 min more on ice. The centrifugation and resuspension were repeated three times and, finally, the pellet was dissolved in 35 mL of 50 mM Tris-HCl (pH 7.6) containing 4 mM EDTA, 8 M urea and 200 mM DTT. The mixture was stirred for 20 min, sonicated for 2 min, stirred on ice for 40 min and centrifuged for 20 min at 15,000 rpm. The pH of the solution was adjusted to 4.5, and it was dialyzed twice against 2 l of 25 mM sodium phosphate buffer (pH 4.5) containing 4 M urea. The precipitate was removed by centrifugation and the solution was passed through a 0.45 μm filter and applied to a SP Sepharose Fast Flow column (GE Healthcare, Chicago, IL, USA) equilibrated with 25 mM sodium phosphate buffer (pH 4.5) containing 4 M urea and 10 mM tris(2-carboxyethyl)phosphine (TCEP). The column was eluted with a linear gradient of NaCl concentration up to 500 mM in the starting buffer. The obtained protein fraction was further purified by reversed-phase HPLC with a gradient of acetonitrile concentration in water in the presence of 0.1% trifluoroacetic acid. The collected protein fraction was freeze-dried and used for refolding.

#### 4.2.3. Refolding

The refolding was performed as described by [22]. In brief, 5 mg of the protein was dissolved in 1 mL of 6 M guanidinium hydrochloride to obtain a concentration of 5 mg/mL. Folding was carried out at 4 °C and initiated by rapid dilution of the protein solution with 50 mL of folding buffer (100 mM Tris, 2.0 M urea, 0.5 M arginine, 4 mM reduced glutathione, 1 mM oxidized glutathione, adjusted to pH 8.0 with conc. HCl). The reaction was left at 4 °C for 3 days after which the mixture was acidified with trifluoroacetic acid to give a pH of ~4 and purified by HPLC on a Jupiter C18 column (10 × 250 mm, Phenomenex, Torrance, CA, USA) with a gradient of acetonitrile concentration from 20 to 50% in 60 min. The purified protein was freeze-dried and used for further study.

#### 4.2.4. Synthesis of sSLURP-1

The synthetic sSLURP-1 was prepared as described in [22].

### 4.3. Analysis of sSLURP-1 and rSLURP-1 Binding to Membrane Preparations of the Torpedo californica and Human α7 nAChRs

In competition experiments with [^125^I]-α-bungarotoxin, different concentrations of rSLURP-1 and sSLURP-1 from the stock solutions (in which the proteins’ concentrations were calculated from the UV spectra, taking into account an extinction coefficient of 3605 at 280 nm and molecular masses of 8975 and 8837 Da, respectively) were preincubated for 3 h at room temperature with nAChRs (human α7 nAChR-expressing GH4C1 cells or *T. californica* electric organ membranes at a final concentration of toxin-binding sites of 0.40 nM and 0.55 nM, respectively), in 50 μL of buffer consisting of 20 mM Tris-HCl and 1 mg/mL BSA, pH 8.0 (binding buffer). Radioiodinated α-bungarotoxin was added to a final concentration of 0.7 nM, and the mixture was incubated for 5 min. Binding was stopped by rapid filtration on GF/C filters (Whatman, Maidstone, UK) pre-soaked in 0.25% polyethylenimine; unbound radioactivity was removed from the filters by washout (3 × 3 mL) with the binding buffer. Non-specific binding was determined in all cases using 3 h of preincubation with 30 μM α-cobratoxin from *Naja kaouthia*.

### 4.4. Two-Electrode Voltage Clamp Analysis of Rat α9α10 nAChR Inhibition by sSLURP-1 and rSLURP-1

Xenopus laevis frogs were fed twice a week and maintained according to supplier recommendations (https://www.enasco.com/page/xen_care, accessed on 1 October 2023). All experiments were carried out in strict accordance with the World Health Organization’s International Guiding Principles for Biomedical Research Involving Animals. The protocol (protocol number: 251/2018 26.02.18) was approved by the Institutional Animal Care and Use Committee based on the Institutional Policy on the Use of Laboratory Animals of the Shemyakin-Ovchinnikov Institute of Bioorganic Chemistry RAS.

Oocytes were removed from mature anesthetized *Xenopus laevis* frogs by dissecting the abdomen and removing necessary amounts of ovarium. Stage V–VI *Xenopus laevis* oocytes were defolliculated with 2 mg/mL collagenase type I (Life Technologies, Camarillo, CA, USA) at room temperature (21–24 °C) for 2 h in Ca^2+^-free Barth’s solution composed of (in mM) 88 NaCl, 1.1 KCl, 2.4 NaHCO_3_, 0.8 MgSO_4_ and 15 HEPES-NaOH at pH 7.6. Oocytes were injected with 9.2 ng of rat nAChR α9 and α10 cRNA (in a ratio of 1:1). Oocytes were incubated at 18 °C for 2–4 days before electrophysiological recordings in Barth’s solution composed of (in mM) 88 NaCl, 1.1 KCl, 2.4 NaHCO_3_, 0.3 Ca(NO_3_)_2_, 0.4 CaCl_2_, 0.8 MgSO_4_ and 15 HEPES-10 NaOH at pH 7.6, supplemented with 40 μg/mL gentamicin and 100 μg/mL ampicillin. Recordings were performed using a turbo TEC-03X amplifier (NPI Electronic, Tamm, Germany) and WinWCP recording software (version 5.7.5, University of Strathclyde, Glasgow, UK). The glass recording electrodes were filled with 3 M KCl and the electrode resistance was 0.1−0.5 MΩ. Membrane potential was clamped at −60 mV. Oocytes were briefly washed with Ba^2+^ Ringer’s solution (Fuchs and Murrow, 1992) composed of (in mM) 115 NaCl, 2.5 KCl, 1.8 BaCl_2_ and 10 HEPES at pH 7.2, followed by three applications of 500 μM acetylcholine (ACh). Washout with Ba^2+^ Ringer’s was performed for 5 min between ACh applications. Oocytes were preincubated with various concentrations of sSLURP1, rSLURP1 and α-conotoxin RgIA for 1 min followed by their co-application with ACh. To induce an ion current, we used a 500 μM ACh concentration. The peak current amplitudes of ACh-induced responses were measured before (ACh alone) and after the preincubation of oocytes with the inhibitors. The ratio between these two measurements was used to assess the activity of the tested compounds. All control experiments were performed on the same day.

Rat α9 and α10 cDNAs were cloned in a pGEMHE vector. Rat α9 and α10 plasmids were linearized using NheI (NEB, Ipswich, MA, USA). mRNAs were transcribed in vitro using a T7 mMESSAGE mMachine™ (Thermo Fisher Scientific, Waltham, MA, USA) and SP6 were prepared using an SP6 mMESSAGE mMACHINE^®^ High Yield Capped RNA Transcription Kit (Thermo Fisher Scientific, Waltham, MA, USA). Transcribed mRNA was polyadenylated using the Poly-A-Tailing Kit (Thermo Fisher Scientific, Waltham, MA, USA). The mRNAs were stored for up to 6 months at −70 °C. Before every use, the degradation levels of mRNAs were checked by gel electrophoresis.

### 4.5. Circular Dichroism (CD) Spectroscopy

CD spectra were recorded on a JASCO J-810 spectropolarimeter (JASCO International Co., Tokyo, Japan) in the range from 190 to 250 nm. The light path length was 0.1 mm. The temperature was changed from 20 to 90 °C in increments of 10 °C. The peptides were dissolved in 10 mM sodium phosphate buffer (pH 7.2) at a concentration of 0.1 mg/mL. Four spectra were averaged for each point. The results were expressed as molar ellipticity, [Θ] (deg × cm^2^ × dmol^−1^), determined as [Θ] = Θ × 100 × MRW/(c × L), where Θ is the measured ellipticity in degrees at a wavelength, MRW is the mean amino acid residue weight, c is the peptide concentration in mg/mL and L is the light path length in centimeters. The instrument was calibrated with (+)-10-camphorsulfonic acid, assuming [Θ]_291_ = 7820 deg × cm^2^ × dmol^−1^ [42]. The analysis of the secondary structure was performed by the CONTIN/LL algorithm (https://sites.google.com/view/sreerama, accessed on 22 June 2021), using the SMP56 protein reference set. NRMSD were used as a statistical estimate of the difference between the experimental spectrum and the theoretical spectrum derived from the obtained composition of the secondary structure. According to [43], NRMSD < 0.1 indicates the high reliability of the calculations.

### 4.6. Raman Spectroscopy of rSLURP-1 and sSLURP-1

Drops of sSLURP-1 and rSLURP-1 solutions of volume 2.5 μL were dried on a calcium fluoride substrate. The protein was predominantly concentrated in the areas at the droplets’ borders (the so-called “coffee ring” effect), which made it possible to record sufficiently intense spectra by focusing on these areas (Appendix A).

Raman spectra measurements were performed using the Raman microscope SENTERRA II (Bruker GmbH, Mannheim, Germany). The exciting laser radiation wavelength was 532 nm. Samples were irradiated with light focused by a 50× objective lens with a 0.65 numerical aperture. Under these conditions, the laser spot on the sample was about 2.5 µm in diameter. The laser power was 12 mW. During the recording of each spectrum, the integration time was 2 sec and averaging over 200 spectra was applied. 

For a comparative analysis of sSLURP-1 and rSLURP-1, the spectrum of each sample was recorded 10 times, and the averaged spectra were compared (a total of 20 spectra were recorded). All spectra were processed by applying (1) baseline correction (concave rubberband correction), (2) normalization (to a peak of 1004 cm^−1^) and (3) smoothing (number of smoothing points = 17) in OPUS 8.2.28 (Bruker Optik GmbH, Ettlingen, Germany).

### 4.7. Computer Modeling of rSLURP-1 and sSLURP-1 Binding to the Torpedo californica and Human α7 nAChRs

Molecular modeling of rSLURP-1’s interaction with orthosteric sites of muscle-type and α7 nAChRs was performed using the ToxDock Rosetta service (https://rosie.graylab.jhu.edu/, accessed on 1 October 2023) [31]. To construct the starting complex structure, PDB coordinates of muscle-type (PDB 6UWZ) and α7 nAChR (PDB 7KOO) structures in complex with α-bungarotoxin [25,44] were utilized. The rSLURP-1 structure (PDB 6ZZF [23]) was aligned with α-bungarotoxin using the PyMOL (TM) Molecular Graphics System, Version 2.6.0a0 (Schrodinger, Inc., New York, NY, USA). After the removal of α-bungarotoxin coordinates, structures were submitted to ToxDock.

Un-guided docking of rSLURP-1 to the α7 nAChR was performed on the HDOCK server [32] using the PDB 7KOO structure for a receptor part and PDB 6ZZF for a ligand molecule. Resulting putative complex structures were analyzed in PyMOL.

## 5. Conclusions

The results obtained show that, between samples of sSLURP-1 and rSLURP-1, the former identical in sequence to the naturally occurring protein and the latter having an additional N-terminal Met residue, there is no great differences either in their experimentally monitored interactions with the *Torpedo californica,* human α7 and rat α9α10 nAChR subtypes or in computer modeling of their binding to the first two targets. In our experiments, we found that both SLURP-1 forms bind to the orthosteric sites of the muscle-type and (although much less efficiently) α7 nAChRs. Although earlier electrophysiological experiments indicated interactions with the allosteric sites in the α7 nAChR for rSLURP-1 and, to a lesser extent, for sSLURP-1 [16,22], the orthosteric sites in the muscle-type nAChR were found in our work to be predominant for both SLURP-1 forms. Computer modeling indicated a similarity in their binding modes to the muscle-type and α7 nAChRs. sSLURP-1’s inhibition of neuronal heteromeric α9α10 nAChRs [22] was shown to be shared by rSLURP-1. It is not yet clear how important the somewhat higher thermostability of sSLURP-1 as found in our work is. Whatever their mode of binding, the analyzed SLURP-1 forms interact with the indicated nAChR subtypes and may find applications in developing drugs acting on these receptors involved in pain, inflammation and neurodegenerative diseases. 

## Figures and Tables

**Figure 1 ijms-24-16950-f001:**
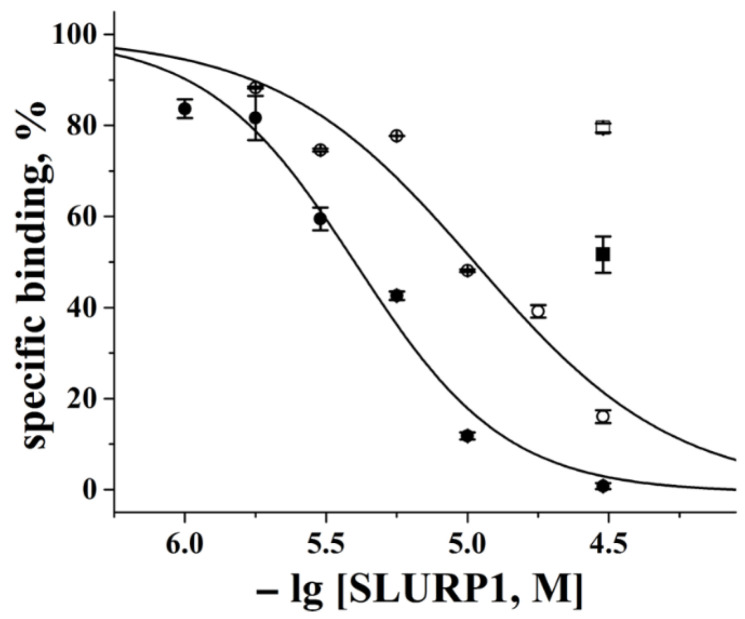
Inhibition of the initial rate of specific [^125^I]-α-bungarotoxin binding to *T*. *californica* (circles) and human α7 (squares) nAChRs by rSLURP-1 (open symbols) and sSLURP-1 (filled symbols). The figure shows the representative dose–response curves of one batch of rSLURP-1 and one batch of sSLURP-1 acting on the *Torpedo* nAChR; the IC_50_ values obtained in this experiment were 10.7 ± 0.9 μM and 4.05 ± 0.25 μM, respectively. Each data point represents the mean ± SEM of 2 replicates.

**Figure 2 ijms-24-16950-f002:**
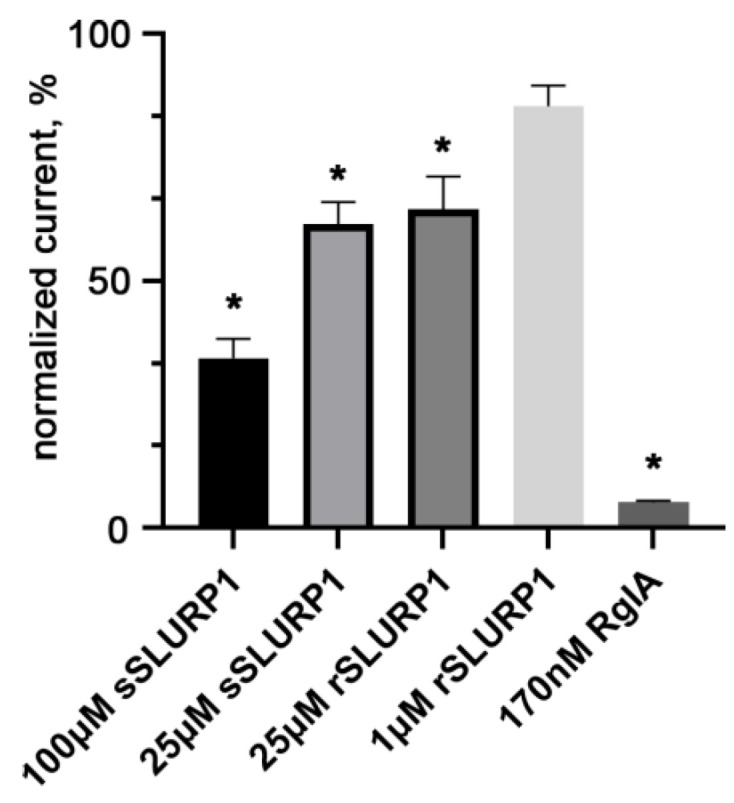
Electrophysiological recordings of sSLURP-1, rSLURP-1 and α-conotoxin RgIA inhibition of acetylcholine (500 μM)-evoked currents in Xenopus laevis oocytes expressing rat α9α10 nAChR. Each plot point represents data obtained from three or four independent experiments (mean ± SEM, vs relative current amplitude in the absence of an inhibitor, Student’s *t*-test, * *p* < 0.05).

**Figure 3 ijms-24-16950-f003:**
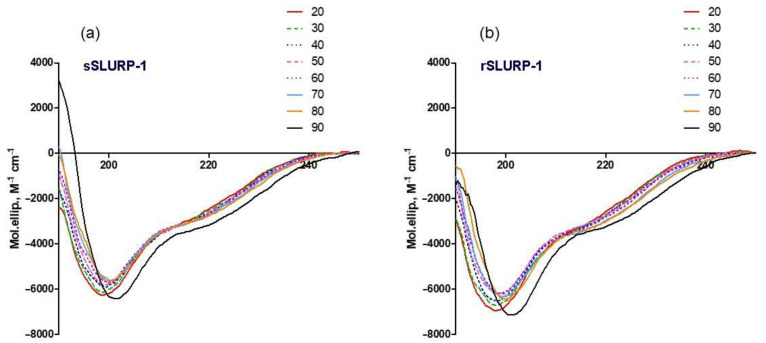
CD spectra of sSLURP-1 (**a**) and rSLURP-1 (**b**) registered at different temperatures.

**Figure 4 ijms-24-16950-f004:**
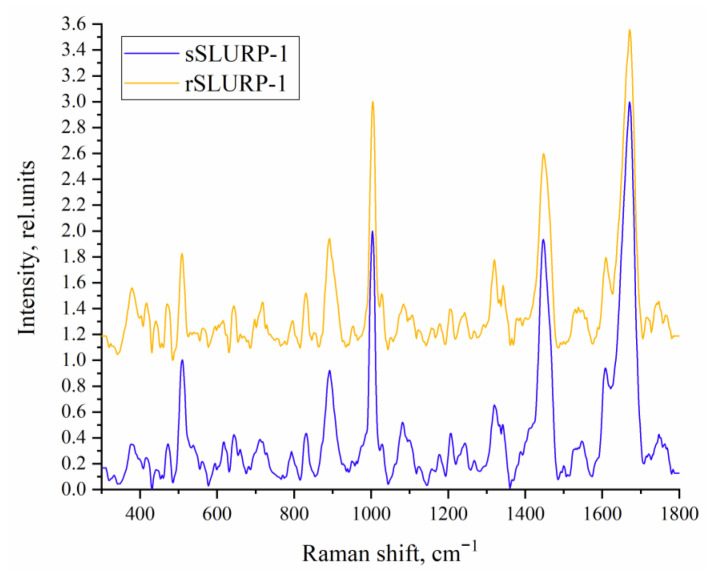
The averaged Raman spectra of sSLURP-1 and rSLURP-1. Each spectrum was obtained by averaging over 5 spectra of the sample, taken at different points in its volume.

**Figure 5 ijms-24-16950-f005:**
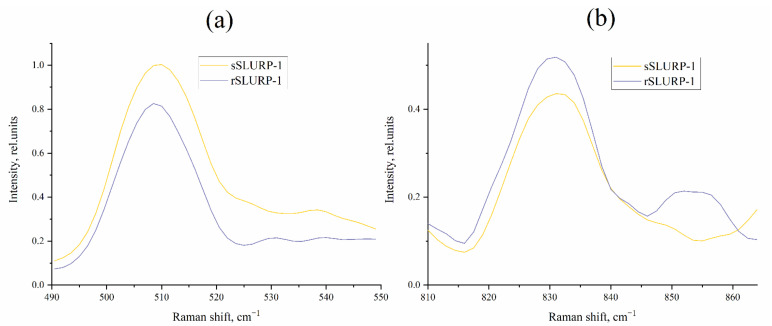
Fragments of the Raman spectra displaying “S–S region” (**a**) and tyrosine doublet region (**b**).

**Figure 6 ijms-24-16950-f006:**
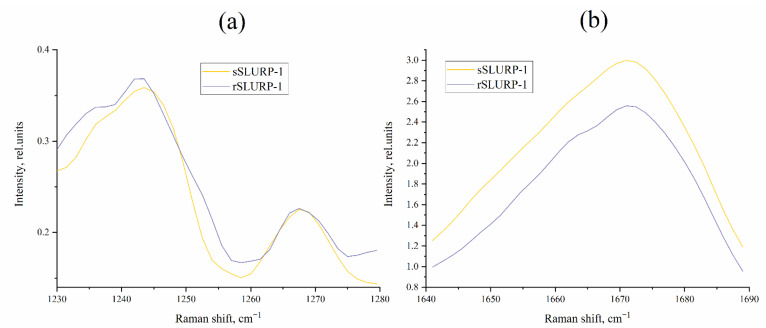
Comparison of the Raman spectra of sSLURP-1 and rSLURP-1 in the region of Amide III (**a**) and Amide I (**b**) bands.

**Figure 7 ijms-24-16950-f007:**
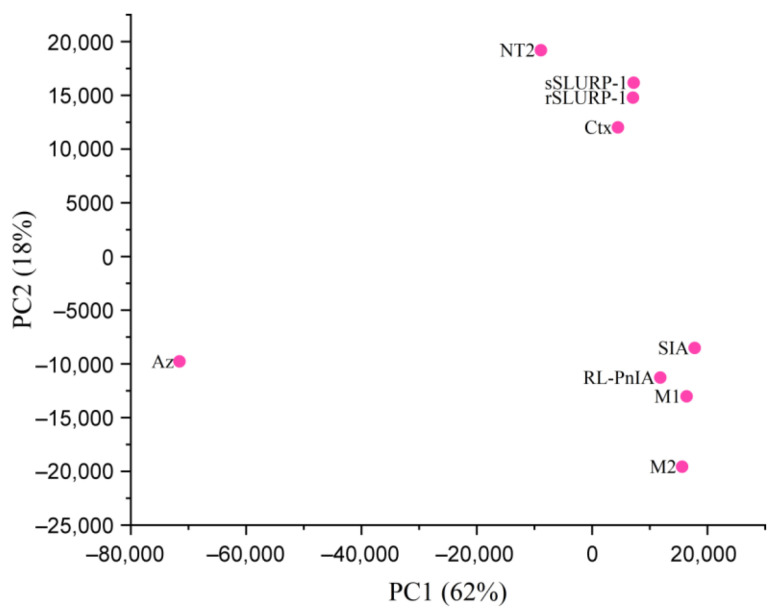
PCA score plot for Raman spectra of nAChR ligands. Percentage of explained variance for each principal component (PC1 and 2) is indicated in brackets.

**Figure 8 ijms-24-16950-f008:**
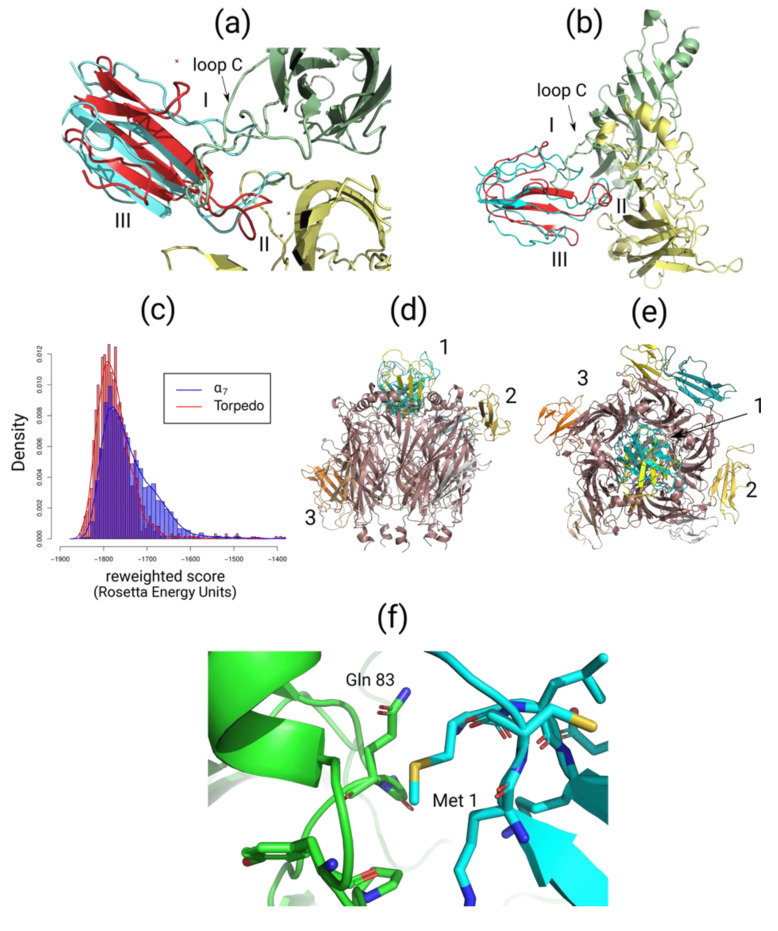
Molecular docking simulation of rSLURP-1 binding to the *T. californica* muscle-type and human α7 nAChRs. (**a**) Top view of putative complex of the muscle-type nAChR (PDB 6UWZ) with α-bungarotoxin (red) from cryo-EM structure with the superimposed rSLURP-1 (light blue). Red cross designates water molecules from the original structure. rSLURP-1 demonstrates a binding pose similar to α-bungarotoxin with loop II of the three-finger protein placed under loop C of the α1 nAChR subunit. (**b**) Docking of rSLURP-1 (light blue) to α7 nAChR orthosteric site (PDB 7KOO, which corresponds to the resting state of the receptor bound to α-bungarotoxin (red)). (**c**) Distributions of Rosetta scoring function for muscle-type and α7 nAChR orthosteric complexes with rSLURP-1. (**d**) Side view of the results of un-guided docking of rSLURP-1 to the extracellular domain of α7 nAChR points out several alternative binding sites outside orthosteric binding pocket. Numbers 1–3 indicate different rSLURP-1 poses. (**e**) Top view of rSLURP-1 poses generated using un-guided docking. Numbers 1–3 indicate different rSLURP-1 poses. (**f**) Close view of the first rSLURP-1 docking pose in α7 nAChR vestibulum site (see (**d**,**e**)). Docking reveals the potential role of Met 1 residue in binding to this allosteric site which might explain some differences between rSLURP-1 and sSLURP-1 in binding to nAChRs.

**Table 1 ijms-24-16950-t001:** Temperature dependencies of secondary structures of sSLURP-1 and rSLURP-1.

NRMSD ^1^		Unordered, %		β-Turn, %		β-Structure, %		α-Helix, %		Temperature, °C
Recombinant	Synthetic	Recombinant	Synthetic	Recombinant	Synthetic	Recombinant	Synthetic	Recombinant	Synthetic	
0.03	0.03	37.2	35.9	22.2	22.2	35.5	36.7	5.1	5.3	20
0.03	0.03	36.9	36.3	22.3	22.1	35.5	36.4	5.2	5.2	30
0.02	0.03	36.1	35.6	22.2	22.1	36.5	36.8	5.3	5.4	40
0.03	0.03	36.1	35.6	22.2	22.1	36.7	37.1	5.1	5.2	50
0.03	0.03	35.9	35.1	22.1	22.0	36.6	37.5	5.4	5.3	60
0.03	0.04	35.9	35.3	22.1	22.0	36.5	37.7	5.5	4.9	70
0.04	0.03	35.0	35.1	22.1	21.9	37.4	37.6	5.5	5.5	80
0.05	0.04	35.3	33.5	22.5	22.2	35.0	38.4	7.2	6.0	90

^1^ normalized root-mean-square deviation.

## Data Availability

All data obtained in this study are contained within the article.

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
