# Peer review of "Comparison of Conformations and Interactions with Nicotinic Acetylcholine Receptors for E. coli-Produced and Synthetic Three-Finger Protein SLURP-1"

_ijms, 2023, doi:10.3390/ijms242316950_

Round 1

Reviewer 1 Report

Comments and Suggestions for Authors

The paper states that the synthetic SLURP-1 (sSLURP-1) has a higher thermostability than the recombinant SLURP-1 (rSLURP-1). However, it later mentions that rSLURP-1 has some tyrosine residues in a hydrophilic environment, indicating less stability. These statements seem contradictory and need to be clarified.

While the study compares the two forms of SLURP-1, it does not compare these forms with other proteins or drugs currently used to target the same receptors. Such a comparison could provide a clearer understanding of the potential advantages or disadvantages of SLURP-1.

The authors should be cautious about generalizing their findings to other Ly6/uPAR proteins. The properties and functions of proteins can vary significantly, even among proteins within the same family. Therefore, the findings related to SLURP-1 may not necessarily apply to other Ly6/uPAR proteins.

Author Response

We are grateful to the Reviewer 1 for the valuable comments which all were addressed and corresponding changes were done in the text.

Comment

The paper states that the synthetic SLURP-1 (sSLURP-1) has a higher thermostability than the recombinant SLURP-1 (rSLURP-1). However, it later mentions that rSLURP-1 has some tyrosine residues in a hydrophilic environment, indicating less stability. These statements seem contradictory and need to be clarified.

Response

We do not see any contradiction: synthetic sSLURP-1 has a higher thermostability and it is recombinant rSLURP-1 which has a tyrosine residue in a hydrophilic environment which may be one of the factors decreasing the thermostability of the latter. However, we are not stating that tyrosine residues are the sole factor explaining the difference between the two SLURP-1 forms in thermostability. Other factors, for example, different geometry of disulfide bonds can affect the thermostability.

Comment

While the study compares the two forms of SLURP-1, it does not compare these forms with other proteins or drugs currently used to target the same receptors. Such a comparison could provide a clearer understanding of the potential advantages or disadvantages of SLURP-1.

Response

We are grateful for this comment and appropriate information is added to the text: historically, α-bungarotoxin from the snake venom was the first three-finger neurotoxin used for isolation and characterization of first muscle-type nAChR. Other three-finger neurotoxins as such or in modified forms are used at present in research on nAChRs, but they all are very toxic. Another modern tool in nAChR studies are α-conotoxins, peptides from Conus marine snails, for which the name also presumes the toxicity. The advantage of such proteins as Lynx1 or SLURP-1, in comparison with α-neurotoxins and α-conotoxins, is their lack of toxicity. Moreover, these proteins are present in the organisms of mammalians and regulate some of important functions which allows to consider them and their derivatives as possible drug leads. The relevant information is added to the text. (The first paragraph of Discussion)

Comment

The authors should be cautious about generalizing their findings to other Ly6/uPAR proteins. The properties and functions of proteins can vary significantly, even among proteins within the same family. Therefore, the findings related to SLURP-1 may not necessarily apply to other Ly6/uPAR proteins.

Response

We agree with this comment: an attempt of generalization to other Ly6/uPAR proteins was made in our first version of this manuscript, but it was criticized by the Reviewer and is not present in this revised manuscript.

Reviewer 2 Report

Comments and Suggestions for Authors

The paper under review addresses the conformational comparison between two forms of the SLURP-1 protein: one expressed in E. coli (recombinant or rSLURP-1) and one synthetically prepared (sSLURP-1). It explores their interactions with two forms of nAChR, specifically the alpha7 nicotinic receptor and the muscle-type receptor from Torpedo californica.

The topic of the article is in line with those of the journal but should be accepted after major revisions.

The main objective of this study becomes evident in the introduction section, where it is explained that the research aims to understand whether the E. coli-expressed protein (rSLURP-1) can serve as a suitable model for studying natural SLURP-1, despite containing an additional methionine residue at the N-terminus compared to the natural protein. While the objective is clear to this reviewer, it is recommended that the rationale for preferring rSLURP-1 over sSLURP-1 be elaborated upon. This reviewer speculates that the preference may stem from the ease of producing recombinant proteins compared to synthetically derived ones, but such a rationale should be explicitly stated in the text for the benefit of non-experts in this field.

The authors employ circular dichroism (CD) and Raman spectroscopy to reveal conformational details of the SLURP-1 proteins. CD is also used to gain insights into protein thermostability. Additionally, competition biochemical assays using radio-labeled natural substrates of the nAChRs have been conducted to gather information about the binding of SLURP-1 to nAChRs, encompassing all the forms examined in this study. Furthermore, computational modeling has been employed to describe the structural aspects of these bindings. The investigation and results are well explained, but this reviewer's main concern pertains to the choice of techniques to compare proteins that are highly similar from a conformational point of view. Indeed, the level of detail provided by CD and Raman could not be suitable in this case. Techniques capable of offering more detailed structural information should be employed because small structural differences can be pivotal in such cases. Therefore, it is recommended that crystallographic or NMR structures be used to draw conclusions about the differences between the two forms of SLURP-1. Regarding the protein-protein modeling study, this reviewer strongly recommends the utilization of Small Angle X-ray Scattering-derived molecular envelopes to validate the docking model. Without such validation, the models obtained through computational simulations will always be open to criticism.

There are also a few minor comments:

  1. In the "Results" section, there are frequent mentions of details related to materials and methods. It is suggested that these details be either moved to the appropriate section or omitted if they are already included in the "Materials and Methods" section.
  2. In Figure 1, the use of black and white colors makes it challenging to discern the error bars. It is advised to modify the colors or symbols for improved clarity.
  3. Please indicate the variable on which NRMSD in Table 1 is calculated.
  4. The passage in the main text from line 136 to 140 is not clear to this reviewer. Please rephrase the sentence, as "Principal Component Analysis of Raman spectra for comparison of these neurotoxins and α-conotoxins interacting with the nAChRs [24]" does not seem to fit the context of the main text.
  5. The passage from line 159 to 161 in the main text is unclear. Please rephrase the sentence, as the usage of "minimal" appears to be inappropriate for the context of the main text.
  6. The use of "Indeed" at line 191 seems out of place in the context of the sentence. Please consider removing it.
  7. Please add a loading plot to Figure 6 and provide commentary, highlighting which regions of the Raman spectrum contribute to distinguishing the structures.
  8. The passage from line 205 to 207 in the main text states that, according to the binding energy from Rosetta, there is a preference of the muscle-type nAChR for rSLURP-1. However, it is not reported respect to what. This reviewer assumes that the comparison is respect to sSLURP-1 but, in such a case, the conclusion should be the opposite, as the biochemical assay using radio-labeled bungarotoxin shows a lower IC50 for sSLURP-1 than rSLURP-1 (slightly lower but lower). Please clarify or rephrase this section.
  9. Please provide a clearer explanation of the color code in the caption for Figure 7. For instance, the reason for using red color for models in 'a' and 'b' should be explained.
  10. Please close the round brackets after "(see reviews [4-6,32])".

These suggestions are made with the aim of improving the clarity and comprehensibility of the paper. 

Author Response

We are grateful to the Reviewer 2 for the valuable comments; all of them were taken into account  and corresponding changes were done in the manuscript.

Comment

The main objective of this study becomes evident in the introduction section, where it is explained that the research aims to understand whether the E. coli-expressed protein (rSLURP-1) can serve as a suitable model for studying natural SLURP-1, despite containing an additional methionine residue at the N-terminus compared to the natural protein. While the objective is clear to this reviewer, it is recommended that the rationale for preferring rSLURP-1 over sSLURP-1 be elaborated upon. This reviewer speculates that the preference may stem from the ease of producing recombinant proteins compared to synthetically derived ones, but such a rationale should be explicitly stated in the text for the benefit of non-experts in this field.

Response

We agree with this absolutely correct remark: there is an obvious fact that a production of such protein as SLURP-1 in E. coli is much easier than the peptide synthesis and we are glad that our comparison demonstrated that the heterologously expressed rSLURP-1 is indeed very close to the sSLURP-1. Our information is relevant to the rSLURP-1 containing only one additional N-terminal Met residue, but we can say nothing about the described in literature rSLURP-1 variants containing quite large fusion portions and an appropriate remark is added to the text (Page 10, paragraph 3).

Comment

The authors employ circular dichroism (CD) and Raman spectroscopy to reveal conformational details of the SLURP-1 proteins. CD is also used to gain insights into protein thermostability. Additionally, competition biochemical assays using radio-labeled natural substrates of the nAChRs have been conducted to gather information about the binding of SLURP-1 to nAChRs, encompassing all the forms examined in this study. Furthermore, computational modeling has been employed to describe the structural aspects of these bindings. The investigation and results are well explained, but this reviewer's main concern pertains to the choice of techniques to compare proteins that are highly similar from a conformational point of view.

Indeed, the level of detail provided by CD and Raman could not be suitable in this case. Techniques capable of offering more detailed structural information should be employed because small structural differences can be pivotal in such cases. Therefore, it is recommended that crystallographic or NMR structures be used to draw conclusions about the differences between the two forms of SLURP-1.

Response

We are grateful to Reviewer who writes that our research and its results are well explained. There is no doubt that crystallographic or NMR structures would give more detailed information about the similarity or differences between the two forms of SLURP-1. However, these are very challenging tasks: still there is no crystal structure of any SLURP-1, moreover, crystallization conditions can differently affect conformations of these proteins. In general, a great similarity of 1H-NMR spectra of sSLURP-1 and rSLURP-1 was demonstrated in [Durek et al. Sci Rep. 2017;7(1):16606]. A further more detailed comparison of the two forms is complicated because of the recently revealed presence of the isomeric rSLURP-1 forms in solution [Paramonov et al. Int J Mol Sci. 2020;21(19):7280] which can be somewhat different for the sSLURP-1. We decided to use two more simple approaches, namely CD and Raman spectroscopy, to assess the similarity of the recombinant and synthetic SLURP-1 forms, that is with the techniques which can be used by other researches for comparing other proteins having not very great structural differences. Our conclusion is that in the case of SLURP-1 the additional N-terminal Met in general can be an acceptable modification. As mentioned above we are not generalizing this conclusion, but in this revised version  added discussion and corresponding references to the publications where the native three-finger protein WTX (weak toxin), having the same disposition of disulfide as SLURP-1 and isolated from the cobra venom) and the E.coli produced rWTX (with one additional N-terminal Met) were compared in their action on the nAChRs and on the muscarinic acetylcholine receptors: these publications showed that depending on the target, there may be common features and some differences between such proteins. (Page 10, the third paragraph). In addition in the revised version of manuscript we added some new electrophysiological data about functional similarity of sSLUPR1 and rSLURP1 as inhibitors of α9α10 nAChR.

Comment

Regarding the protein-protein modeling study, this reviewer strongly recommends the utilization of Small Angle X-ray Scattering-derived molecular envelopes to validate the docking model. Without such validation, the models obtained through computational simulations will always be open to criticism.

Response

We agree with the reviewer that the structures obtained by computational simulations need confirmation by some structural methods. However, this is not easy task in the case of model of SLURP-1 bound to nAChR which is a membrane bound protein. Small Angle X-ray Scattering is very good method to study water soluble of detergent solubilized protein. Certainly, nAChR can be solubilized, but the ligand binding characteristics of solubilized form are different from those of the native membrane bound receptor. There are no data about SLURP-1 interaction with solubilized nAChR, but recently, it was shown that SLURP-1 did not interact with water soluble acetylcholine-binding protein which is a good model of nAChR extracellular domain containing ligand-binding sites [Durek et al. Sci Rep. 2017;7(1):16606]. So, before using Small Angle X-ray Scattering it is necessary to elaborate conditions if any for SLURP-1 binding to the solubilized nAChR. However, this task is beyond the scopes of this work.

There are also a few minor comments:

Comment

  1. In the "Results" section, there are frequent mentions of details related to materials and methods. It is suggested that these details be either moved to the appropriate section or omitted if they are already included in the "Materials and Methods" section.

Response

The description of some experimental details was removed from the Results section.

Comment

  1. In Figure 1, the use of black and white colors makes it challenging to discern the error bars. It is advised to modify the colors or symbols for improved clarity.

Response

The symbols were diminished to make the error bars visible.

Comment

  1. Please indicate the variable on which NRMSD in Table 1 is calculated.

Response

Normalized root mean square deviations (NRMSD) were used as a statistical estimate of the difference between the experimental spectrum and the theoretical spectrum derived from the obtained composition of the secondary structure. According to [Kelly et al. Biochim Biophys Acta. 2005;1751(2):119-39] NRMSD < 0.1 indicates the high reliability of the calculations. This explanation was added to the Section 4.4. Circular dichroism (CD) spectroscopy.

Comment

  1. The passage in the main text from line 136 to 140 is not clear to this reviewer. Please rephrase the sentence, as "Principal Component Analysis of Raman spectra for comparison of these neurotoxins and α-conotoxins interacting with the nAChRs [24]" does not seem to fit the context of the main text.

Response

The text was corrected to make the passage clearer.

Comment

  1. The passage from line 159 to 161 in the main text is unclear. Please rephrase the sentence, as the usage of "minimal" appears to be inappropriate for the context of the main text.

Response

The text has been corrected.

Comment

  1. The use of "Indeed" at line 191 seems out of place in the context of the sentence. Please consider removing it.

Response

The paragraph was rewritten to make it clearer.

Comment

  1. Please add a loading plot to Figure 6 and provide commentary, highlighting which regions of the Raman spectrum contribute to distinguishing the structures.

Response

The loading plot has been added to the supplementary materials (Figure S2). Various regions (marker bands), characterizing both vibrations of protein backbone and side-chains, contribute to the distinguishing of various toxins and it is hardly possible to highlight any specific region (or several regions). This explanation was added to the text. (Page 7, the first paragraph)

Comment

  1. The passage from line 205 to 207 in the main text states that, according to the binding energy from Rosetta, there is a preference of the muscle-type nAChR for rSLURP-1. However, it is not reported respect to what. This reviewer assumes that the comparison is respect to sSLURP-1 but, in such a case, the conclusion should be the opposite, as the biochemical assay using radio-labeled bungarotoxin shows a lower IC50 for sSLURP-1 than rSLURP-1 (slightly lower but lower). Please clarify or rephrase this section.

Response

Binding energy estimated by Rosetta scoring function shows preferential interaction of rSLURP-1 with muscle-type nAChR compared to α7 receptor. This phrase was corrected.

Comment

  1. Please provide a clearer explanation of the color code in the caption for Figure 7. For instance, the reason for using red color for models in 'a' and 'b' should be explained.

Response

α-Bungarotoxin molecule is shown in red. This was corrected.

Comment

  1. Please close the round brackets after "(see reviews [4-6,32])".

Response

This was corrected.

Round 2

Reviewer 2 Report

Comments and Suggestions for Authors

-

Author Response

We thank the Reviewer 2 for the comments. Minor revisions have been done to the manuscript according to the Editor recommendations.